# Fracture-GS: Dynamic Fracture Simulation with Physics-Integrated Gaussian Splatting

**Xiaogang Wang**[1,*]**, Hongyu Wu**[2,*]**, Wenfeng Song**[3]**, Kai Xu**[4]

[1]College of Computer and Information Science, Southwest University
[2]State Key Laboratory of Virtual Reality Technology and Systems, QRI & SCSE, Beihang University
[3]School of Computer, Beijing Information Science and Technology University
[4]Institute of AI for Industries, Chinese Academy of Sciences

## Abstract

This paper presents a unified framework for simulating and visualizing dynamic fracture phenomena in extreme mechanical collisions using multi-view image inputs. While existing methods primarily address elastic deformations at contact surfaces, they fail to capture the complex physics of extreme collisions, often producing non-physical artifacts and material adhesion at fracture interfaces. Our approach integrates two key innovations: (1) an enhanced Collision Material Point Method (Collision-MPM) with momentum-conserving interface forces derived from normalized mass distributions, which effectively eliminates unphysical adhesion in fractured solids; and (2) a fracture-aware 3D Gaussian continuum representation that enables physically plausible rendering without post-processing. The framework operates through three main stages: First, performing implicit reconstruction of collision objects from multi-view images while sampling both surface and internal particles and simultaneously learning surface particle Gaussian properties via splatting; Second, high-fidelity collision resolution using our improved Collision-MPM formulation; Third, dynamic fracture tracking with Gaussian attribute optimization for fracture surfaces rendering. Through comprehensive testing, our framework demonstrates significant improvements over existing methods in handling diverse scenarios, including homogeneous materials, heterogeneous composites, and complex multi-body collisions. The results confirm superior physical accuracy, while maintaining computational efficiency for rendering.

## 1 Introduction

Dynamic fracture simulation stands at the intersection of computational physics and computer graphics, enabling realistic modeling of material failure across diverse applications. While physics-based methods like the Material Point Method (MPM) have advanced significantly since their introduction (Stomakhin et al., 2013), critical gaps remain in handling extreme mechanical collisions and achieving seamless simulation-to-rendering pipelines.

Recent advances in explicit scene representation, particularly 3D Gaussian splatting (Kerbl et al., 2023), have revolutionized real-time rendering capabilities. Building upon this foundation, several studies have successfully integrated physical simulation with Gaussian representations, including (Borycki et al., 2024; Cai et al., 2024; Feng et al., 2024; Xie et al., 2024; Zhang et al., 2025). Among these, (Xie et al., 2024) established a significant milestone by coupling 3D Gaussians with MPM simulations, demonstrating remarkable adaptability across various material types. Parallel developments include (Cai et al., 2024)'s physics parameter estimation through Gaussian differentiability and (Borycki et al., 2024)'s GASP framework for point-wise physical attribute embedding. While these approaches have made substantial progress in bridging the simulation-rendering gap, their applicability remains constrained to moderate mechanical conditions and specific material classes.

A critical limitation emerges when addressing extreme mechanical collisions, such as the high-energy fragmentation observed in brittle materials (Wolper et al., 2019). Current methodologies face two fundamental challenges in these scenarios: (1) unphysical adhesion artifacts in MPM simulations , and (2) inadequate fracture surface representation for rendering. These challenges become

---

*Corresponding authors: `wangxiaogang@swu.edu.cn`, `whyvrlab@buaa.edu.cn`

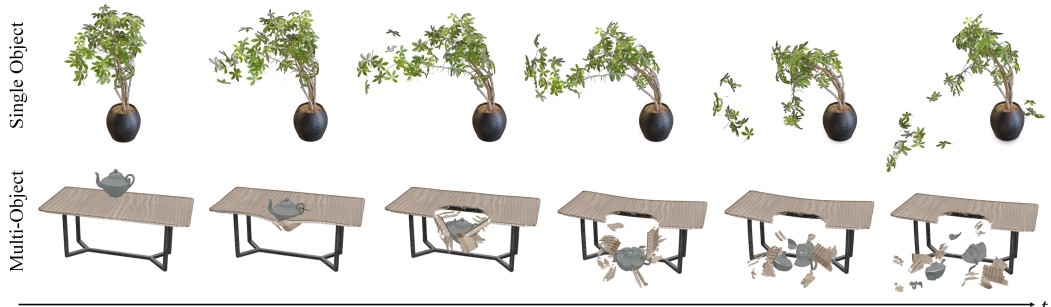

Figure 1: Two extreme mechanical collision scenarios are simulated using our proposed framework: (1) single-object impacts against wind, and (2) multi-object collision interactions with complex fracture dynamics. Notably, the flowerpot object comprises heterogeneous materials (with distinct properties for leaves, stems, and the pot itself), and the table object consists of legs and tabletop, while other teapot is modeled as homogeneous materials.

particularly apparent in high-energy impact scenarios like brittle material fragmentation or multi-body collisions.

To address these limitations, we introduce Fracture-GS, a unified framework for simulating and visualizing dynamic fracture phenomena in extreme mechanical collisions using multi-view image inputs. First, the Signed Distance Function (SDF) of the object is constructed from multi-view images to implicitly represent the volumetric geometry of the object, followed by sampling both surface and internal particles within the SDF-constrained domain to ensure spatial coherence; Meanwhile, surface particles learn Gaussian attributes using isotropic kernels. Subsequently, we proposed an enhanced collision-MPM, which is used to perform extreme collisions between multiple objects with dynamic fracture. It can effectively alleviate the non-physical adhesion phenomenon caused by MLS-MPM. Then, all fracture particles are tracked through a hardening-aware tracking criterion defined by (Wolper et al., 2019). Finally, based on the tracking fracture particles, we can efffciently regenerate their Gaussian attributes through the proposed fracture particles Gaussian optimization strategy, enabling high-quality rendering, as shown in Figure 1.

In summary, our key contributions are:

- A unified physics-rendering framework that combines our enhanced Collision-MPM with fracture-aware 3D Gaussian representations, enabling high-fidelity simulation and visualization of extreme mechanical collisions across diverse materials.

- An improved Collision-MPM formulation that introduces momentum-conserving interface forces derived from normalized mass distributions, effectively eliminating the non-physical adhesion artifacts prevalent in conventional MPM simulations.

- A dynamic fracture tracking and rendering pipeline that identifies fracture surfaces through hardening parameter and reconstructs Gaussian attributes via minimal-volume enclosing ellipsoid (MVEE) optimization.

## 2 PRELIMINARY AND RELATED WORK

### 2.1 3D GAUSSIAN SPLATTING AND DYNAMIC SCENE RECONSTRUCTION

3D Gaussian Splatting (3DGS) is a fast neural rendering method that primarily represents volumetric scenes using a collection of anisotropic 3D Gaussian kernels. Each Gaussian kernel is defined by a set of differentiable parameters $\{x_p, \sigma_p, A_p, c_p\}$, where $x_p$ denotes the spatial position, $\sigma_p$ represents opacity, $A_p$ is the covariance matrix, and $c_p$ is the view-dependent color function. The covariance matrix $A_p$ can be further decomposed into scale $s_p$ and rotation $r_p$ components, which control the spatial distribution and orientation of the Gaussian kernel, respectively. The color function $c_p$ achieves view-dependent characteristics through spherical harmonics, capturing complex lighting and material effects. The rendering process involves projecting (splatting) the 3D Gaussian kernels onto a 2D image plane, incorporating viewpoint transformation, opacity blending, and depth sorting. The final color of the $i$-th pixel is computed using the following formula:

$$C_i = \sum_{k \in I} \sigma_k c_k (d_i) \prod_{j=1}^{k-1} (1 - \sigma_j) \tag{1}$$

Here, $I$ stands for the set of Gaussian kernels, $\sigma_k$ and $\sigma_j$ represent the opacity of the $k$-th and $j$-th Gaussian kernels, respectively. $d_i$ means the viewing direction from the camera to the $i$-th pixel, and $c_k(d_i)$ indicates the color of the Gaussian kernels in the viewing direction $d_i$. Since 3D Gaussian explicitly represents the scene, its learning and rendering speeds are generally faster than those of 3D reconstruction methods based on NeRF. As a result, a wide range of applications based on 3D Gaussian have emerged.

Dynamic 3D scene reconstruction has long been a challenging problem, aiming to reconstruct dynamic scenes from various representations such as videos and images. The introduction of Neural Radiance Fields (NeRF) has significantly advanced this field, leading to a series of dynamic scene reconstruction methods that build upon NeRF's framework. These methods focus on addressing challenges such as non-ideal input conditions, including sparse views and motion blur, to enhance reconstruction quality. Notable works in this domain include (Pumarola et al., 2021), which handles dynamic scenes by incorporating temporal information, and (Fang et al., 2022), which improves efficiency through time-aware neural voxels. More recently, (Song et al., 2023) proposed a streamable dynamic scene representation that decomposes neural radiance fields for efficient reconstruction and rendering.

Another approach to scene representation is based on 3D Gaussian Splats (3DGS), which explicitly represents 3D scenes using a set of Gaussian kernels and achieves fast rendering through Gaussian splatting. Due to its high efficiency and interpretability, several dynamic scene reconstruction methods based on 3DGS have been proposed, such as (Lin et al., 2024; Huang et al., 2024; Wu et al., 2024; Yang et al., 2024; Sun et al., 2024; Zhang et al., 2024; Dahmani et al., 2024). Among these, (Lin et al., 2024) incorporates DDDM (Deformable Dynamic Model) into the optimization process of Gaussians, eliminating the need to reconstruct Gaussian for each frame and directly guiding Gaussian deformation using DDDM. On the other hand, (Huang et al., 2024) learns control points for Gaussian and uses a small number of control points to govern the motion of the entire Gaussian set, achieving efficient reconstruction of dynamic motion processes. (Dahmani et al., 2024) divides the dynamic sequence into different windows based on the motion number, and train dynamic Gaussian models for different windows, together with different canonical spaces and deformation fields.

## 2.2 MATERIAL POINT METHOD AND PHYSICS-BASED GAUSSIAN APPROACHES

The Material Point Method (MPM) is a numerical approach based on a hybrid Eulerian-Lagrangian framework, used to solve governing equations in continuum mechanics and facilitate bidirectional information transfer between particles and grids. Let $p \in \{P_a, P_b\}$ denote the classification criterion for distinguishing particle subsets. Following established methodologies, the process adheres to the MLS-MPM framework (Hu et al., 2018), which is divided into the following three stages:

- **Particle-to-Grid (P2G) Stage:** The particle mass $m_p$ and velocity $v_p$ are interpolated to neighboring grid nodes through a weighted projection scheme governed by basis functions:

$$m_i^n = \sum_p w_{ip}^n m_p, \tag{2}$$

$$m_i^n v_i^n = \sum_p w_{ip}^n m_p \left( v_p^n + C_p^n(x_i - x_p^n) \right), \tag{3}$$

  where $w_{ip}^n$ represents the interpolation kernel (e.g., quadratic B-spline) evaluated at particle position $x_p$ for grid node $i$, $P$ is the set of active particles, and $m_i$ and $v_i$ are the aggregated mass and velocity at grid node $i$, respectively.

- **Grid Update Stage:** Grid velocities are advanced by solving the discrete momentum conservation equations through an explicit forward Euler integration scheme:

$$m_i^n(v_i^{n+1} - v_i^n) = -\Delta t \cdot f_i^* + \Delta t \cdot f_i^{ext}, \tag{4}$$

$$f_i^* = \sum_p \frac{4}{\Delta x^2} V_p w_{ip}^n \sigma_p^n(x_i^n - x_p^n), \tag{5}$$

where $\Delta x$ denotes the grid size, $f_i^*$ is the grid force calculated from the particle volume $V_p$, Cauchy stress $\sigma_p^n$, and positions $x_i^n$ and $x_p^n$, and $f_i^{ext}$ is the external force (typically gravity).

- **Grid-to-Particle (G2P) Stage:** The updated grid velocities $v_i^{n+1}$ are mapped back to Lagrangian particles to update their kinematic states for the next time step $n+1$. This transfer is achieved through interpolation:

$$v_p^{n+1} = \sum_i w_{ip}^n v_i^{n+1}, \tag{6}$$

$$C_p^{n+1} = \frac{4}{\Delta x^2} \sum_i w_{ip}^n v_i^{n+1} (x_i^n - x_p^n)^T. \tag{7}$$

After obtaining the updated particle velocities, the particle positions are advanced through an explicit time integration scheme:

$$x_p^{n+1} = x_p^n + \Delta t v_p^{n+1}. \tag{8}$$

The Material Point Method (MPM) has emerged as a powerful tool for simulating complex physical phenomena, combining the advantages of both Lagrangian and Eulerian approaches. Recent advancements in MPM simulations have significantly expanded its applications and performance optimization. (Hu et al., 2018) introduces a variant of the MPM based on Moving Least Squares (MLS), referred to as MLS-MPM, for simulating complex physical phenomena involving displacement discontinuities and bidirectional rigid-body coupling. By incorporating the Compatible Particle-In-Cell (CPIC) algorithm, this method enables the handling of discontinuities in material points, infinitely thin boundaries, and bidirectional coupling with rigid bodies. As a result, it is capable of simulating material cutting, dynamic open boundaries, and interactions between rigid and deformable bodies. (Wolper et al., 2019) integrates a phase field into the MPM to develop a crack-tracking approach known as PFF-MPM. Additionally, it proposes an incompressible plastic flow rule that maintains constant volume during plastic stress projection. Meanwhile, (Fang et al., 2020) proposed a novel framework for fluid-solid coupling using IQ-MPM. This method combines a "ghost matrix" operator splitting scheme with weak-form governing equations to achieve stable and efficient coupling under the CFL time step constraint. It supports discrete consistency with hybrid Lagrangian-Eulerian solvers and uses an interface quadrature (IQ) technique to handle free-slip boundaries, avoiding the "stickiness" issues in traditional MPM implementations. Moreover, the effectiveness of employing GPUs to enhance the computational efficiency of MPM implementations has been well-documented in several studies, such as (Gao et al., 2018; Hu et al., 2019; Qiu et al., 2023). However, these methods all require post-processing of the simulation results to achieve high-quality rendering effects.

The integration of Gaussian-based techniques into physical simulations leverages the rendering efficiency of Gaussian, thereby eliminating the need for post-processing with dedicated rendering engines such as Houdini after the completion of the physical simulation, and directly yielding results with Gaussian splatting. Specifically, (Xie et al., 2024) incorporates Gaussian kernels into the dynamic simulation process of MPM, enabling continuum mechanics simulations based on 3D Gaussian kernels and achieving real-time rendering in simple scene simulations. (Zhang et al., 2025) uses diffusion on images to obtain prior motion videos of objects, simulates the motion process based on MPM, leverages the differentiability of MPM to learn the material field in specified regions, and completes the forward simulation process to generate dynamic videos. (Cai et al., 2024) aims to guide the learning of physical properties of objects using 3D Gaussian splats (3DGS). It first reconstructs the static Gaussians in the initial state and learns the deformation models of Gaussians based on dynamic inputs, optimizing the initial velocities and physical parameters (Young's modulus and Poisson's ratio) of the static Gaussians through differentiable MPM. (Tan et al., 2024) generates high-quality, physics-based videos from a single image. However, none of these methods address the extreme cases where objects undergo fragmentation, which is the primary focus of this work.

## 3  METHOD

In this section, we introduce Fracture-GS, a unified framework for simulating and visualizing dynamic fracture phenomena in extreme mechanical collisions using multi-view image inputs. As illustrated in Figure 2, we first reconstruct the geometry of colliding objects using multi-view images combined with existing implicit 3D reconstruction algorithms (Xiao et al., 2024), followed by sampling both surface and internal particles within the SDF-constrained domain to ensure spatial

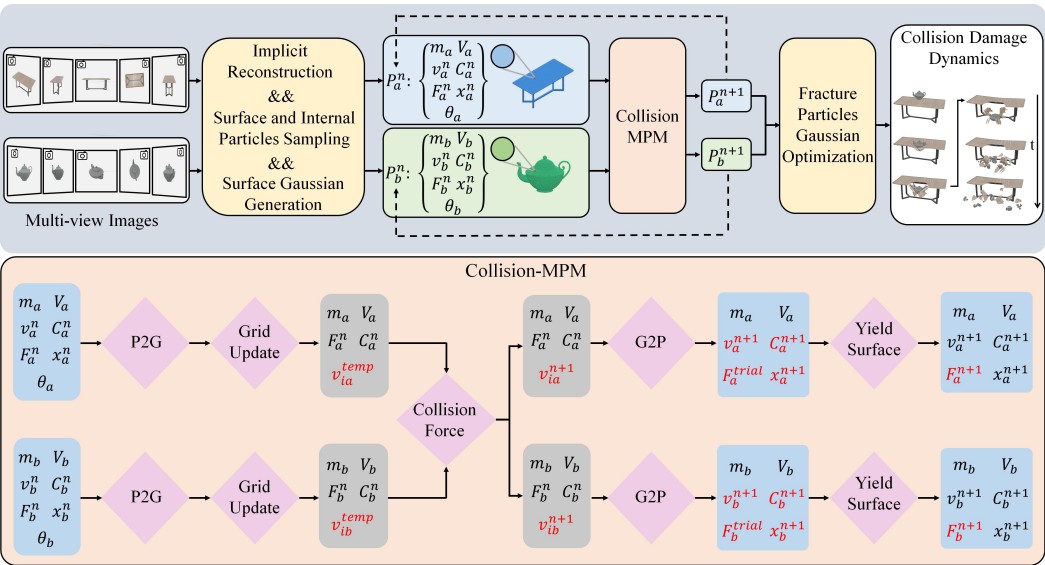

Figure 2: Pipeline. The object is implicitly reconstructed from multi-view images, followed by sampling both surface and internal particles. Surface particles learn Gaussian attributes using isotropic kernels. Next, the sampled particles undergo extreme mechanical collision simulation with dynamic fracture using our enhanced **Collision-MPM**. Finally, fracture particles are tracked and their Gaussian attributes are optimized through our proposed **Fracture Particle Gaussian Optimization** strategy, enabling high-quality rendering of the simulation results. For **Collision-MPM**, the key parameters are highlighted in red. The yield surface determines whether a particle enters the plastic region, triggering a return mapping procedure to project stress back to the yield surface and update the particle's deformation gradient. Parameters in the blue bottom plate are computed in the Lagrangian coordinate system, while those in the gray bottom plate are computed in the Eulerian coordinate system.

coherence; Subsequently, surface particles Gaussian kernels are trained using 3D Gaussian Splatting from input images. Then, we proposed an enhanced collision-MPM, which is used to perform extreme collisions between multiple objects with dynamic fracture. It can effectively alleviate the non-physical adhesion phenomenon caused by MLS-MPM. Finally, to enhance the visual realism of the mechanical simulation, all fracture particles are tracked through a hardening-aware tracking criterion defined by (Wolper et al., 2019), based on the tracking fracture particles (as shown in Figure 3 (right)), we can efffciently regenerate their Gaussian attributes through the proposed fracture particles Gaussian optimization strategy, enabling high-quality rendering.

The all particle attributes of the colliding objects $P_a : \{m_a, V_a, C_a, F_a, v_a, x_a, \theta_a\}$ and $P_b : \{m_b, V_b, C_b, F_b, v_b, x_b, \theta_b\}$ include mass ($m$), volume ($V$), deformation gradient ($F$), velocity gradient ($C$), velocity ($v$), position ($x$), and elastoplastic parameters ($\theta$), where $\theta$ comprises Young's modulus ($E$), Poisson's ratio ($\gamma$), hardening tracking parameters ($\alpha$), cohesion coefficient ($\beta$), and hardening factor ($\xi$).

## 3.1 COLLISION-MPM

Although the Material Point Method (MPM) itself can solve the problem of collisions between objects, its performance has limitations. For instance, in fluid-solid coupling simulations, fluid particles may unnaturally adhere to solid surfaces—a phenomenon similarly observed in multi-body fracture simulations, where fragmented solids exhibit unphysical adhesion to neighboring objects. To mitigate these artifacts, we propose a Collision-MPM framework, integrating the collision force (Yan et al., 2018) into the MPM framework. This method, originally designed to prevent fluid-solid interpenetration, ensures collision realism across material interfaces.

In our **Collision-MPM** framework, the particle information of two objects is independently transferred to the grid. First, after completing the P2G and Grid Update processes, the mass distributions $\hat{n}_{ia}$ and $\hat{n}_{ib}$ of object particles $P_a$ and $P_b$ on grid node $G_i$ are calculated. Simultaneously, the grid velocities $v_i^{n+1}$ are replaced with $v_i^{temp}$. Subsequently, the interface direction tendencies $n_{ia}$ and

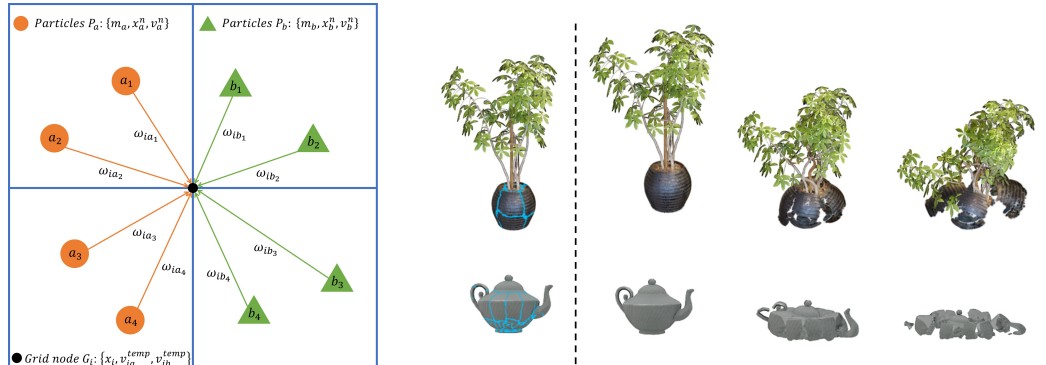

Figure 3: (left) Illustration of the mass distributions of particles $P_a$ and $P_b$ at grid node $G_i$. (right) Tracking fracture particles through hardening factor $\alpha$.

$n_{ib}$ at grid node $G_i$ for object particles $P_a$ and $P_b$ are determined:

$$\hat{n}_{ia} = \frac{\sum_{p_a} m_a \nabla \omega_{ia}(x_i - x_a^n)}{\left\| \sum_{p_a} m_a \nabla \omega_{ia}(x_i - x_a^n) \right\|} \tag{9}$$

$$\hat{n}_{ib} = \frac{\sum_{p_b} m_b \nabla \omega_{ib}(x_i - x_b^n)}{\left\| \sum_{p_b} m_b \nabla \omega_{ib}(x_i - x_b^n) \right\|} \tag{10}$$

As shown in Figure 3 (left), $\hat{n}_{ia}$ and $\hat{n}_{ib}$ illustrate the mass distributions of particles $P_a$ and $P_b$ on grid node $G_i$. We utilize these distributions to compute the interface direction tendency, which can also be interpreted as the tendency of the contact surface's normal direction:

$$n_{ia} = -n_{ib} = \frac{\hat{n}_{ia} - \hat{n}_{ib}}{\|\hat{n}_{ia} - \hat{n}_{ib}\|} \tag{11}$$

The computation of collision forces is conditionally activated based on relative velocity analysis at material interfaces. Specifically, collision forces are generated at grid node $G_i$ only if the inequality $(v_{ia}^{temp} - v_{ib}^{temp}) \cdot n_{ia} > 0$ is satisfied, where $v_{ia}^{temp}$ and $v_{ib}^{temp}$ denote the velocities of $P_a$ and $P_b$ particles on grid node $G_i$ after the grid update. The collision forces are derived from the momentum conservation principle during the collision process:

$$f_i^c = \frac{p_{ia}^{temp} m_{ia}^n - p_{ib}^{temp} m_{ib}^n}{(m_{ia}^n + m_{ib}^n)\Delta t} \tag{12}$$

$$f_{ia}^c = -f_{ib}^c = \mu(f_i^c \cdot n_{ib}) n_{ib} \tag{13}$$

Here, $f_i^c$ represents the collision force on grid node $i$, $m_{ia}^n$ and $m_{ib}^n$ denote the mass contributions of $P_a$ and $P_b$ particles on grid node $G_i$, respectively. $p_{ia}^{temp} = m_{ia}^n v_{ia}^{temp}$ and $p_{ib}^{temp} = m_{ib}^n v_{ib}^{temp}$ represent the momenta of $P_a$ and $P_b$ particles on grid node $G_i$, respectively. $\mu$ is a constant controlling the magnitude of the collision force. $f_{ia, collision}$ and $f_{ib, collision}$ denote the collision forces acting on $P_a$ and $P_b$ particles on grid node $G_i$. After computing the collision forces, they are integrated into the grid velocity update:

$$v_{ia}^{n+1} = v_{ia}^{temp} + \frac{f_{ia}^c}{m_{ia}^n}\Delta t, \quad v_{ib}^{n+1} = v_{ib}^{temp} + \frac{f_{ib}^c}{m_{ib}^n}\Delta t \tag{14}$$

where $v_{ia}^{n+1}$ and $v_{ib}^{n+1}$ represent the updated velocities of $P_a$ and $P_b$ particles at grid node $G_i$, respectively.

## 3.2 Fracture Particles Gaussian Optimization (FPGO).

### 3.2.1 Continuum Mechanics and Constitutive Model.

To simulate the dynamic behavior of an elastic-plastic object, it is essential to solve the conservation equations for momentum and mass:

$$\rho \frac{Dv}{Dt} = \nabla \cdot \sigma + f, \quad \frac{D\rho}{Dt} + \rho \nabla \cdot v = 0 \tag{15}$$

Here, $\rho$ denotes density, $v$ represents the velocity field, and $f$ is an external force. The Cauchy stress tensor, denoted by $\sigma$, is given by: $\sigma = \frac{1}{det(F)} \frac{\partial \psi(F^E)}{\partial F} F^{E^T}$, where $\psi(F)$ is the strain energy density function (or constitutive model), which describes the relationship between stress and strain in a material. The total deformation gradient, $F$, is decomposed into elastic and plastic components: $F = F^E F^P$, enabling the simulation of plastic deformation. In this work, we use the NACC model in (Wolper et al., 2019). It extends CCC (Coherent Cam Clay) model with non-associated flow rules to better simulate plastic deformation while maintaining volume during plastic projection, and it introduced four plastic parameters $\alpha, \beta, \xi, M$ to control the simulation effect of the plastic model.

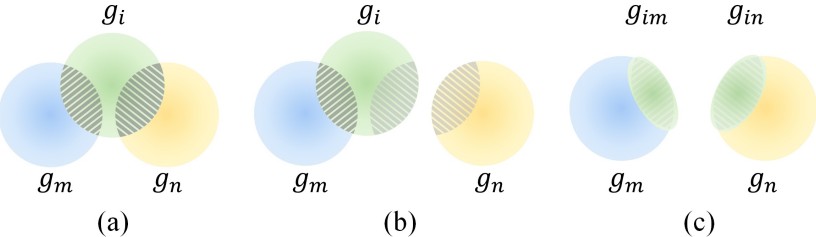

Figure 4: **Illustration of Fracture Particles Gaussian Optimization**

### 3.2.2 FRACTURE TRACKING AND RENDERING

The enhanced Collision-MPM framework successfully simulates extreme mechanical collisions with dynamic fracture phenomena. While one might intuitively consider directly rendering using the initially learned Gaussian attributes of each surface particles, this naive approach leads to significant non-physical artifacts at fracture interfaces (highlighted by red boxes in Figure 5).

The underlying mechanism of these artifacts can be explained as follows: As shown in Figure 4, consider three Gaussian particles $\{g_i, g_m, g_n\}$ in initial configuration. When fracture occurs at particle $g_i$, causing displacement relative to $g_m$ (Figure 4(b)), the increased interparticle distance reduces or eliminates the Gaussian overlap region between $g_i$ and $g_m$, thereby disrupting the continuity of the rendering field and generating visual artifacts (As demonstrated in Figure 4(a) and (b)).

To address this challenge, we propose a novel fracture-aware Gaussian attribute optimization strategy consisting of four key components. First, leveraging the hardening parameter $\alpha$ from the NACC constitutive model (Wolper et al., 2019), we dynamically identify fractured particles at each timestep (visualized as green particles in Figure 4). Taking particle $g_i$ as an example (Figure 4), our tracking begins when its hardening parameter exceeds the hardening parameter threshold $\alpha$.

For each identified fractured particle $g_i$, we perform neighborhood analysis within a radius $d_c$ to locate adjacent intact particles $\{g_m, g_n\}$. This adaptive search range ensures proper coverage of potential interaction zones while maintaining computational efficiency.

The core optimization involves Gaussian cloning and attribute reconstruction. We first compute the minimal-volume enclosing ellipsoid (MVEE) for the overlap regions between $g_i$ and its neighbors $g_m, g_n$, generating two new Gaussian particles $g_{im}$ and $g_{in}$ to replace the original $g_i$. The attribute assignment follows two principles: (1) optical properties including opacity $\alpha$ and color $c$ are directly inherited from $g_i$ to maintain visual consistency; (2) spatial parameters are recomputed through:

$$\{\mu_{new}, \Sigma_{new}\} = \text{MVEE}(\Omega_{cross}(g_i, g_j)) \tag{16}$$

where $\Omega_{cross}$ denotes the original overlap region (Implementation details are provided in Appendix A.4.).

During final rendering, we implement an occlusion-aware sampling scheme: if a pixel's rendering path contains multiple optimized particles ($g_{im}, g_{in}$), only the nearest particle contributes to shading. This prevents overcounting while preserving physical correctness. As demonstrated in Fig 1, our approach generates physically plausible transitional particles that maintain both visual continuity across fracture surfaces and mechanical accuracy in collision regions.

# 4 EXPERIMENTS

## 4.1 EXPERIMENTAL DATA AND PHYSICAL PARAMETER SETTINGS

To comprehensively validate the effectiveness of our experiments, we selected three objects for collision (More experimental results are provided in the supplementary material): Ficus, Teapot and Table. Among these, the Ficus plant and the table are heterogeneous material objects. Specifically, the Ficus plant consists of leaves, branches, and a ceramic pot, each made of different materials, while the table has a tabletop and legs constructed from distinct materials. The teapot is homogeneous in material composition. The physical parameters for all objects are detailed in Appendix A.1. For specific implementation details, please refer to A.2. Additional experimental results are provided in A.5 and Appendix A.6.

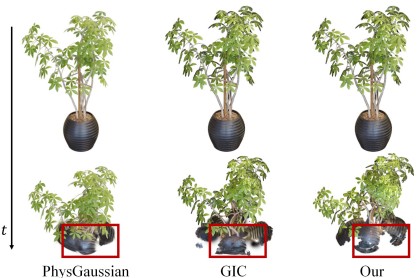

| Method | PSNR ↑ | LPIPS ↓ | FID ↓ | FSF ↑ |
|---|---|---|---|---|
| PhysGaussian | 17.2 | 0.43 | 120.92 | 1.5 |
| GIC | 16.8 | 0.45 | 129.33 | 1.2 |
| Ours (w/o FPGO) | 17.0 | 0.45 | 123.63 | 1.5 |
| Ours (w/o C-MPM) | 20.6 | 0.35 | 93.05 | 3.1 |
| Ours | **21.1** | **0.29** | **90.75** | **3.5** |

Figure 5: Integrated comparison showing both qualitative visualizations (left) and quantitative metrics (right), with our method achieving the best results.

## 4.2 COMPARISON WITH STATE-OF-THE-ART METHODS

We compare our method with two state-of-the-art Gaussian splatting based simulation frameworks. First, **PhysGaussian** (Xie et al., 2024) is a physics-integrated Gaussian framework that simulates and renders mechanical behaviors under external forces from multi-view inputs. Second, **GIC** (Cai et al., 2024), originally designed for material property estimation from videos, is adapted for comparison by utilizing only its forward simulation component with given material parameters, analogous to PhysGaussian. All methods employ identical initialization conditions to ensure fair comparison: the same static 3D Gaussian reconstruction pipeline following GIC's methodology; the same NACC constitutive model was adopted for physical simulation, but the above two comparison methods did not include FPGO and used the MLS-MPM; and identical initial conditions and material parameters. We use PSNR (Hore & Ziou, 2010), LPIPS (Zhang et al., 2018) , and FID (Heusel et al., 2017) as primary metrics to evaluate reconstruction quality. Due to the absence of ground truth for dynamic fracture sequences, we employ a self-referencing evaluation scheme using established image quality metrics. The specific implementation details of our self-referencing metric calculation are provided in Appendix A.3.

**User Study.** We also conducted a human evaluation to assess simulation fidelity, following methods from prior work (Liu et al., 2025; Wei et al., 2024). Ten participants with varying experience in simulation and vision rated to **Fracture Simulation Fidelity (FSF)**, checking if it was realistic and as expected. Rendered videos of simulations were presented in random order, with participants rating each on a five-point scale (1 = poor, 5 = excellent). Mean scores appear in Figure 5 (right).

**Results.** Quantitative and qualitative results in Figure 1 and 5. Our method significantly outperforms other methods in GS-based simulation. Competing methods often produce artifacts due to inadequate or neglected handling of fracture surfaces and collision adhesion, which degrades simulation quality. Although our approach excels in simulation, the object's FID is high due to reliance on relies on training view interpolation for Gaussian restoration and cannot perform Gaussian reconstruction on hidden areas. Future work will explore 3D AI-based texture generative inpainting to improve this.

**Effect of Collision-MPM (C-MPM) and Fracture Particles Gaussian Optimization (FPGO).** To validate C-MPM effectiveness, we conduct ablation studies comparing against the conventional MLS-MPM approach. As demonstrated in Figure 6, the right column reveals significant

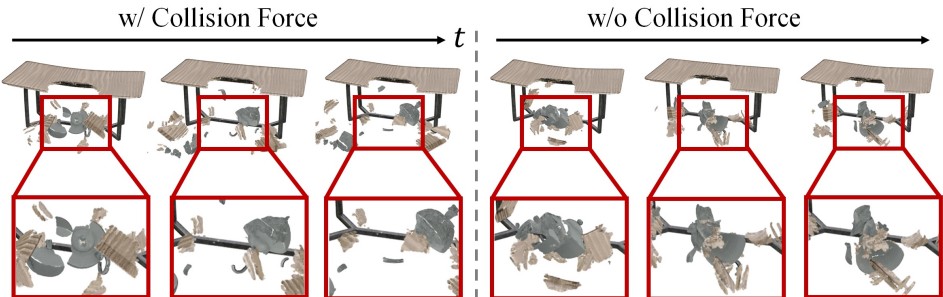

Figure 6: **Collision-MPM** effectively resolves the non-physical adhesion artifacts in multi-body collisions. As demonstrated in the red box regions, wood fragments from the fractured table exhibit natural separation behavior rather than adhering unnaturally to the teapot surface.

non-physical adhesion artifacts (highlighted in red boxes) in MLS-MPM simulations, whereas our method successfully eliminates these unrealistic phenomena through momentum-conserving interface forces derived from normalized mass distributions. Regarding the FPGO module, quantitative experiments in Figure 5 demonstrate its critical advantages. Both baseline methods - PhyGaussian and GIC - directly render using initially learned Gaussian attributes, resulting in various visual artifacts at fracture interfaces. In contrast, our dynamic attribute optimization strategy significantly enhances visual realism by reconstructing Gaussian properties through MVEE-based interpolation.

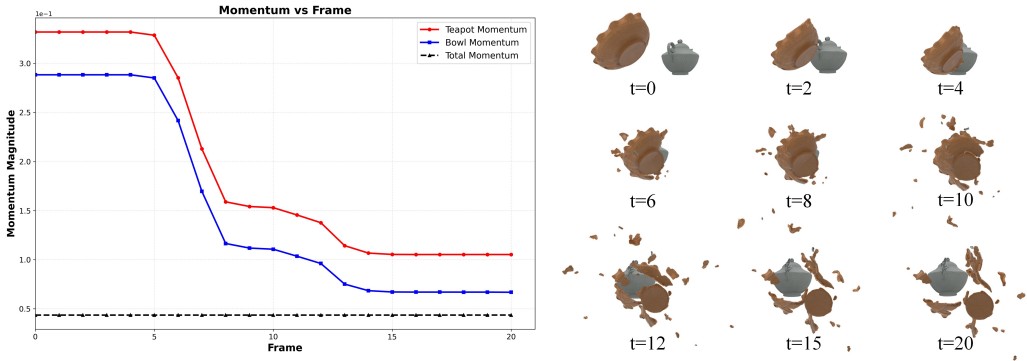

Figure 7: Momentum conservation during a collision between two objects. The total momentum (black curve) remains constant throughout the simulation, demonstrating strict adherence to the conservation law, while the individual momenta of the objects exchange during the impact.

**Momentum Conservation Validation.** To rigorously validate that our simulation framework strictly adheres to the law of momentum conservation, we designed a controlled experiment. This experiment involves two objects—a teapot and a bowl—propelled towards each other with initial velocities in a environment free from external influences such as gravity and friction. The figure 7 tracks the system's evolution over time ($t = 0$ to $t = 20$). It visually demonstrates that while individual momenta change during the collision interval ($t \approx 5$ to $t \approx 15$), the "Total Momentum" curve remains constant in both magnitude and direction throughout the entire sequence. This constant total momentum can be clearly demonstrated from the figure that our simulation method accurately maintains the total linear momentum of the system.

**Energy Stability Analysis.** To quantitatively validate the concern regarding non-physical energy growth and numerical instability, we conduct a thorough energy evolution analysis throughout a representative simulation involving the collision and fracture of a teapot on a table surface. As illustrated in Figure 8, we track the kinetic, elastic, and gravitational potential energy components for both the teapot and the table individually, as well as for the combined system over the first 50 frames. The results demonstrate that the total energy of the system remains strictly bounded and does not exhibit any anomalous increase. Energy is transferred in a physically consistent manner: kinetic energy converts into elastic deformation energy upon impact, and part of it is dissipated through

fracture processes, while gravitational potential energy varies accordingly with object height. The smooth transitions and the absence of energy blow-up confirm that our method inherently prevents non-physical energy accumulation. This energy behavior serves as an effective unit test, validating that our simulation not only captures complex dynamic and fracture phenomena but also maintains numerical stability under severe contact and deformation conditions.

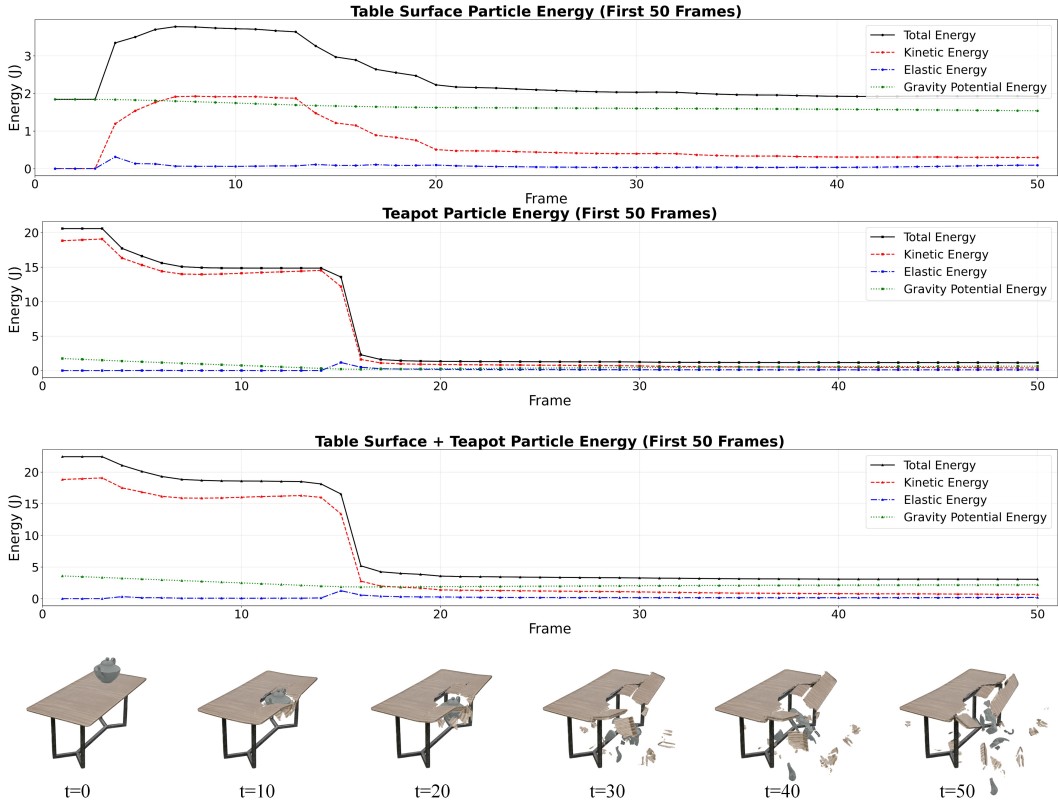

Figure 8: Energy evolution and qualitative visualization during the teapot-table collision and fracture simulation. The top and middle rows plot the kinetic, elastic, and gravitational potential energy components for the table surface, the teapot, and the combined system over the first 50 frames, showing bounded total energy without non-physical growth. The bottom row provides corresponding qualitative visualizations at key frames (t=0,10,20,30,40,50), depicting the physical progression of the collision and fracture process that correlates with the energy transitions observed in the graphs.

## 5 CONCLUSION

This paper presents a unified framework for simulating and rendering extreme mechanical collisions with dynamic fracture effects. The framework demonstrates robust performance across various scenarios, including high-velocity impacts and heterogeneous material fractures. Qualitative and quantitative evaluations show significant improvements over existing methods in both physical accuracy and rendering quality.

Current limitations include high computational demands restricting real-time performance in complex scenes, the need for manual parameter setting, and shortcomings in the evaluation approach—such as indirect quantitative metrics for this task—all of which will be addressed in future work: implementing GPU optimization and adaptive time-stepping for faster computation, developing learning-based methods for automatic parameter estimation, and establishing specialized evaluation metrics to enhance the framework's practicality in virtual prototyping and visual effects production.

## ACKNOWLEDGEMENTS

This work was supported in part by the Chongqing Natural Science Foundation (CSTB2024NSCQ-MSX1026), in part by the Fundamental Research Funds for the Central Universities (No. SWU-KT25012), in part by the Key R&D Program of Shandong Province under grant 2025TSGC-CZZB0165, and in part by the NSFC (62572062).

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

## A  APPENDIX

### A.1  MATERIAL PARAMETERS

Table 1 summarizes the material parameters employed in our collision simulations, including Young's modulus ($E$), Poisson's ratio ($\nu$), density, and non-associated flow rule parameters ($\alpha, \beta, \xi, M$). These physically-based values were assigned to each object component to validate our framework's capability in handling diverse material properties. The selected parameters reflect realistic material contrasts, enabling quantitative analysis of how mechanical properties—such as the stiffness variation between flexible leaves and rigid branches—influence fracture propagation patterns across all experiments.

Table 1: Material Parameters

| Scene | $E$ (MPa) | $\nu$ | Density | NACC ($\alpha, \beta, \xi, M$) |
|---|---|---|---|---|
| Bowl | $5\times10^4$ | 0.46 | 2 | 0.98, 0.5, 1, 2.36 |
| Ficus leaf | $8\times10^4$ | 0.39 | 0.6 | 0.94, 2, 3, 2.36 |
| Ficus branch | $1\times10^6$ | 0.39 | 5 | 0.94, 2, 3, 2.36 |
| Ficus pot | $2\times10^4$ | 0.39 | 2 | 0.98, 0.5, 2, 2.36 |
| Teapot | $5\times10^5$ | 0.46 | 5 | 0.98, 0.5, 1, 2.36 |
| Table top | $1.5\times10^4$ | 0.39 | 1 | 0.99, 0.5, 1, 2.36 |
| Table leg | $1\times10^8$ | 0.39 | 1000 | 0.94, 2, 3, 2.36 |

### A.2  EXPERIMENTS DETAIL

Here, we provide additional details regarding the experiments. These include collisions between single object and rigid surface, collisions among multiple objects, as well as experiments on the interactions between individual objects and rigid surface under varying physical parameters. We also present an ablation study on the impact of fracture particle tracking.

**Input Resolution & Sampling:** The input resolution starts at $1024 \times 1024$. For each collision object, we sample 200,000 surface points and 100,000 interior points.

**Simulation Setup:** Building upon Warp, the simulation is executed on an 18-core Intel Xeon Gold 5220 CPU and an NVIDIA GeForce RTX 3090 GPU, achieving 100-frame sequences for each collisions scene.

**Implementation Details.** Following established practice in GIC (Cai et al., 2024), our simulation pipeline begins with isotropic Gaussian reconstruction for visual surface representation. For physical discretization, we employ an SDF-based voxelization strategy: after constructing a volumetric grid and identifying interior voxels via SDF filtering, we perform uniform random sampling **inside each interior voxel** using a controllable density parameter $N_v$ (samples per voxel). This preprocessing step decouples visual quality from simulation discretization while ensuring physically plausible material sampling. To evaluate the influence of the particle sampling density $N_v$ on the simulation results, we set multiple sets of internal sampling parameters ($N_v = 0, 5, 10, 15, 20, 50, 100, 200$), and the visualization results are shown in Figure 12.

### A.3  METRIC COMPUTATION FOR FRACTURE ASSESSMENT

To quantitatively evaluate the visual plausibility of fracture propagation in the absence of ground truth dynamic sequences, we designed a self-referencing assessment protocol using three established image quality metrics: PSNR, LPIPS, and FID. Our evaluation strategy focuses on measuring how faithfully each method maintains visual continuity during the fracture process. For each fracture event, we identify the last frame before fracture initiation (detected via the hardening parameter $\alpha$) as the reference frame $I_{\text{ref}}$, representing the intact object's appearance. We then analyze the subsequent 5 frames where fracture propagation becomes fully visible, ensuring this window captures

critical fracture dynamics while maintaining visual comparability to the reference. The core principle underlying our metric design is that high-quality fracture rendering should appear as a coherent extension of the original material. Accordingly, we compute PSNR to measure pixel-level consistency with the pre-fracture state, LPIPS to assess perceptual similarity to the intact appearance, and FID to evaluate distributional similarity to pre-fracture rendering. Methods that produce visual artifacts or unnatural fracture surfaces consequently exhibit large deviations from $I_{\text{ref}}$, resulting in degraded metric scores that reflect their reduced visual plausibility.

It should be noted, however, that this represents a pragmatic compromise in the absence of more principled evaluation methodologies. We regard the development of dedicated metrics for physics-based rendering as an important direction for future work.

## A.4 MINIMAL-VOLUME ENCLOSING ELLIPSOID (MVEE)

### A.4.1 PROBLEM DEFINITION

Given two spheres in 3D space with centers $\mathbf{C}_1$, $\mathbf{C}_2$ and radii $r_1, r_2$, we aim to compute the **minimal-volume enclosing ellipsoid (MVEE)** of their intersection region.

**Step 1: Intersection Conditions and Geometric Parameters**

  **Intersection Criteria.** The spheres intersect if:

$$|r_1 - r_2| < d < r_1 + r_2, \quad \text{where} \quad d = \|\mathbf{C}_1 - \mathbf{C}_2\|$$

If $d \geq r_1 + r_2$, the spheres are disjoint; if $d \leq |r_1 - r_2|$, one sphere is entirely contained within the other.

  **Key Geometric Properties.** The intersection region is a *lens-shaped* volume bounded by two spherical caps. Its properties include:

- **Symmetry axis**: The line connecting $\mathbf{C}_1$ and $\mathbf{C}_2$ (unit vector $\mathbf{u} = \frac{\mathbf{C}_2 - \mathbf{C}_1}{d}$).
- **Maximal width**: Perpendicular to $\mathbf{u}$, determined by the radius of the circle of intersection.

**Step 2: Analytical Estimation of the Enclosing Ellipsoid**

  **Ellipsoid Center.** The center $\mathbf{c}$ of the MVEE is approximated as a weighted midpoint along the symmetry axis:

$$\mathbf{c} = \mathbf{C}_1 + \left(\frac{r_1}{r_1 + r_2}\right)(\mathbf{C}_2 - \mathbf{C}_1)$$

This heuristic prioritizes the larger sphere's influence.

  **Ellipsoid Axes.** The ellipsoid has three principal axes:

1. **Major axis (aligned with $\mathbf{u}$)**:

$$a = r_1 + r_2 - d$$

2. **Minor axes (perpendicular to $\mathbf{u}$)**: Lengths $b = c$, given by the radius of the intersection circle:

$$b = c = \sqrt{r_1^2 - \left(\frac{d^2 + r_1^2 - r_2^2}{2d}\right)^2}$$

  **Orientation Matrix.** The ellipsoid's rotation matrix $\mathbf{R}$ is constructed from the orthonormal basis:

$$\mathbf{R} = [\mathbf{u} \quad \mathbf{v} \quad \mathbf{w}], \quad \text{where} \quad \mathbf{v} \perp \mathbf{u}, \ \mathbf{w} = \mathbf{u} \times \mathbf{v}$$

## A.5 COMPUTATIONAL EFFICIENCY

We analyze the computational overhead of our fracture tracking mechanism by comparing the per-frame rendering time for three core scenarios in main paper, with and without this feature enabled. As detailed in Table 2, the incorporation of fracture tracking introduces a moderate and consistent computational cost, increasing rendering time by approximately 20-80% across the scenes. This overhead is attributed to the additional steps of dynamically updating the Gaussian splatting model to reflect new fracture surfaces and collisions in each frame.

## A.6 ABLATION STUDIES

**Effect of Fracture Particle Tracking and Gaussian Generation.** During the fracture process, some internal particles become exposed and visible. To achieve higher visual realism, we need to quickly locate these particles and generate corresponding Gaussian visual attributes. Based on the

Table 2: Per-frame Gaussian rendering time.

| Scene | w/o fracture tracking | w/ fracture tracking |
|---|---|---|
| Ficus | 49.19 ms | 62.78 ms |
| Teapot | 6.13 ms | 10.87 ms |
| Teapot & Table | 27.15 ms | 29.85 ms |

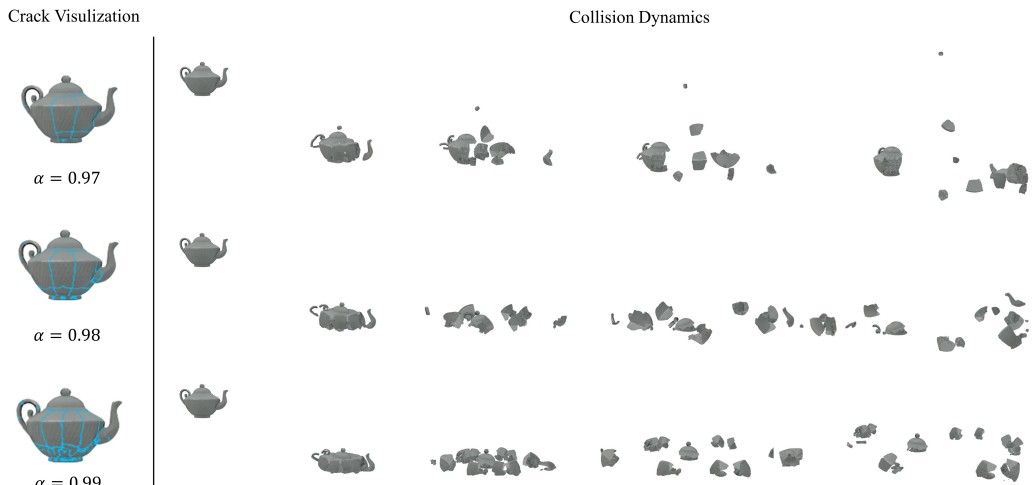

Figure 9: Collisions with three distinct initial hardening factors.

hardening tracking parameters ($\alpha$), we can efficiently track relevant internal particles on the fracture surface and rapidly generate their Gaussian visual attributes through interpolation with neighboring external surface particles, enabling high-quality rendering. Additionally, a large number of isolated particles may be generated during collisions. We utilize the fracture particle tracking mechanism to exclude these fine particles from Gaussian rendering. This selective exclusion ensures that only particles meeting specific material integrity criteria participate in the final rendering, thereby maintaining the physical accuracy while improving computational efficiency.

**Single-Object Analysis.** To systematically evaluate our method's sensitivity to material properties, we conduct extensive collision experiments on individual objects with varying plasticity parameters, as shown in Figure 9. Our results demonstrate that increasing the initial hardening factor systematically enhances object fragmentation and promotes more extensive crack propagation, validating our method's ability to capture material-dependent fracture behaviors.

**Multi-Object Interactions.** To assess performance in complex scenarios, we execute multiple challenging multi-body interaction experiments, as shown in Figure 10. These include bowl-teapot collisions, bowl-table impacts, and ficus-table interactions. These experiments demonstrate our framework's robustness in handling heterogeneous material compositions and complex contact dynamics across diverse object categories.

**Real-world Data Validation.** To further validate the generalizability of our approach, we extend our evaluation to real-world data from the DTU dataset. As shown in Figure 11 and 12, our pipeline successfully performs Gaussian reconstruction and initialization from real images, followed by physically plausible fracture simulation. These results demonstrate our method's robustness when applied to real-world captured data, confirming its practical applicability beyond synthetic environments.

**Analysis of sampling point density.** To assess the influence of particle sampling density on simulation outcomes, we conducted a systematic sweep of internal sampling parameters ($N_v = 0, 5, 10, 15, 20, 50, 100, 200$). Using the highest density configuration ($N_v = 200$) as a convergence reference, we evaluated trajectory deviations across all sampling conditions (Figure 12 and Table 3). The results reveal a fundamental characteristic of MPM simulations: particle sampling density intrinsically influences the resolved mechanical response. As shown in our quantitative analysis and

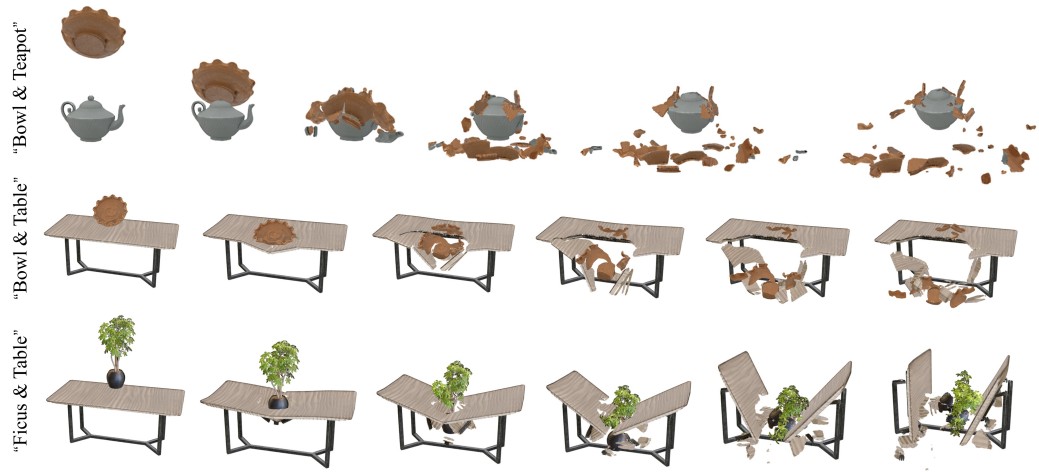

Figure 10: Collisions between different objects: "Bowl & Teapot", "Bowl & Table", "Ficus & Table".

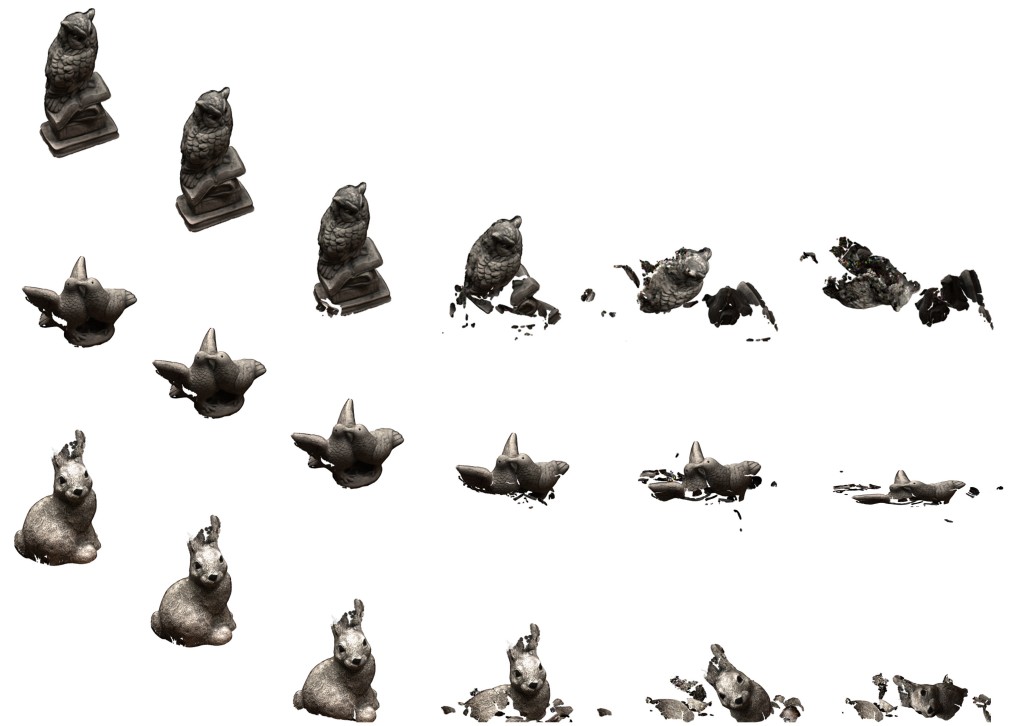

Figure 11: Real-world data validation of our method: results for three different real-world objects.

qualitative visualizations (Figure 12 and Table 3), variations in sampling density naturally lead to differences in fracture patterns and collision dynamics—this is an inherent property of particle-based methods where discretization density affects solution convergence.

| Interior Point Density | X-coordinate Error | Y-coordinate Error | Z-coordinate Error |
|:---:|:---:|:---:|:---:|
| 0 | 0.1998 | 0.1479 | 0.0535 |
| 5 | 0.1936 | 0.1433 | 0.0474 |
| 10 | 0.1884 | 0.1360 | 0.0445 |
| 15 | 0.1772 | 0.1279 | 0.0435 |
| 20 | 0.1786 | 0.1224 | 0.0412 |
| 50 | 0.1713 | 0.1191 | 0.0337 |
| 100 | 0.0935 | 0.0749 | 0.0315 |

Table 3: Coordinate errors at different interior point densities.

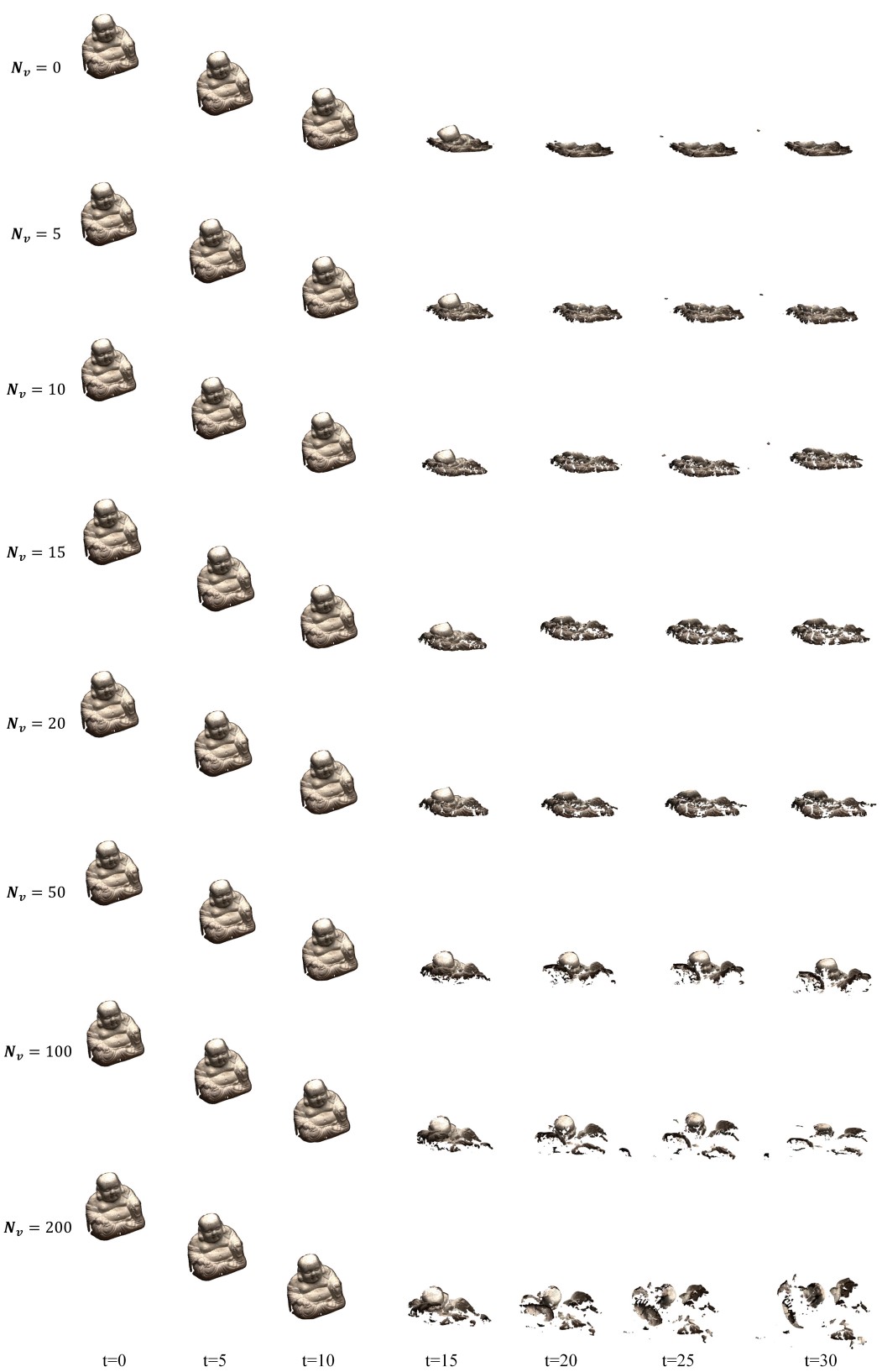

Figure 12: Visual comparison of simulation and rendering results under different internal sampling densities ($N_v$) across three representative timesteps.

