# OpenReview forum: "Fracture-GS: Dynamic Fracture Simulation with Physics-Integrated Gaussian Splatting"
_ICLR.cc/2026/Conference — ICLR 2026 Poster_

### Official Review · Reviewer_yz5d · 2025-10-29

**Soundness:** 3
**Presentation:** 3
**Contribution:** 2
**Rating:** 2
**Confidence:** 5

**Summary:**

The authors propose an extension to the PhysGaussian framework to handle fractures. In their simulation, they use separate velocity fields for different objects and apply collision forces to the nodes that are commonly affected. This contact model can eliminate the sticky artifacts when two objects collide. The NACC plasticity model is employed to simulate fracture behavior. However, this improvement of MPM simulation is novel. The contact model seems identical to the one in [1]. For rendering, the hardening factor is used to track crack propagation and to split Gaussian kernels accordingly. This handling is kind of novel and clever.

[1] Wretborn, J., Armiento, R. and Museth, K., 2017. Animation of crack propagation by means of an extended multi-body solver for the material point method. Computers & Graphics, 69, pp.131-139.

**Strengths:**

- The handling of fractures in GS-based MPM simulation is indeed a neat solution.

**Weaknesses:**

- The contact model is claimed to be one of contributions. However, the model seems identical to the one in [1]. A clarification on difference and novelty is needed.

- The experiment is insufficient. Only 3 examples are provided for comparison and the user study. The conclusion is not significant enough.

- The simulation scenes are overly idealized. The presented examples do not fully demonstrate the strength of Gaussian splatting in reconstructing real-world data. A traditional MPM simulation and rendering pipeline should be able to handle better.

- A limitation is that the framework seems quite bound to NACC or similar plasticity model that has hardening mechanism. However, MPM supports many commonly used plasticity models.

- Some unprofessional typos:
  - Line 256: Figure 7?
  - Line 344: The deformation gradient is not additive.
  - Line 369: "and" is in math mode
  - Line 425: zhang2018unreasonable?

[1] Wretborn, J., Armiento, R. and Museth, K., 2017. Animation of crack propagation by means of an extended multi-body solver for the material point method. Computers & Graphics, 69, pp.131-139.

**Questions:**

- How image metrics are computed? I assume there is no ground truth for simulated results. If only the reconstruction error is compared, it is not fair. I would expect that all method can start from the same initial Gaussian Splatting representation? Are there any modifications to GS reconstruction to increase the image metrics?

- How is $\mu$ in Eq.13 set? I would expect that a too small value can cause penetration and a too large value can cause simulation explosion?

- How to make sure the total energy (kinematics energy + elasticity energy + gravity potential) does not increase to cause instability.  Unit tests on energy should be included.

- The plant in Fig. 5 should not be plastic. How different materials are assigned?

---

> ### Author Response · Authors · 2025-11-27
>
> **Response to: "How image metrics are computed? I assume there is no ground truth for simulated results. If only the reconstruction error is compared, it is not fair. I would expect that all method can start from the same initial Gaussian Splatting representation? Are there any modifications to GS reconstruction to increase the image metrics?"**
>
> We thank the reviewer for this profound question regarding our evaluation methodology. We fully acknowledge the challenge of assessing dynamic fracture rendering without ground truth video. Our evaluation protocol was carefully designed to address this specific challenge through a combination of **targeted quantitative metrics** and **human perceptual evaluation**.
>
> ### 1. Quantitative Metrics for Fracture Assessment
>
> Since no ground truth dynamic sequence exists, we designed our quantitative metrics (PSNR, LPIPS, FID) to specifically measure the **visual plausibility of fracture propagation**:
>
> - **Reference Frame Selection**: For each fracture event, we use the **last frame before fracture** (identified via hardening parameter $\alpha$) as reference frame $I_{\text{ref}}$. This represents the intact object's appearance immediately before fracture.
>
> - **Test Frame Selection**: We select the **subsequent 5 frames** where fracture propagates and becomes fully visible. This window captures critical fracture dynamics while maintaining visual comparability to the reference.
>
> - **Metric Calculation**: We compute:
>   - **PSNR**: Measures pixel-level consistency with pre-fracture state
>   - **LPIPS**: Assesses perceptual similarity to intact appearance
>   - **FID**: Evaluates distributional similarity to pre-fracture rendering
>
> The core principle is: **high-quality fracture should appear as a coherent extension of the original material**. Methods producing artifacts show large deviations from $I_{\text{ref}}$, resulting in poor scores.
>
> ### 2. Human Evaluation as Ultimate Validation
>
> We recognize quantitative metrics alone are insufficient for this novel task. Therefore, we introduced:
>
> **Fracture Simulation Fidelity (FSF)** - A user study where participants rate:
> - Physical plausibility of fracture patterns
> - Visual realism of fracture surfaces
> - Overall simulation credibility
>
> We consider human perceptual evaluation the **most reliable validator** when computational metrics are imperfect.
>
> ### 3. Identical Initial Setup
>
> All three methods (Ours, PhysGaussian, and GIC) employ:
> - **The same static 3D Gaussian reconstruction pipeline** following GIC's methodology
> - **Identical isotropic Gaussians with scale regularization** for detailed appearance representation
> - **The same NACC constitutive model** for physical simulation
> - **Identical initial conditions and parameters**
>
> ### 4. Core Differentiators
>
> The performance differences stem entirely from our two novel contributions that are absent in baseline methods:
>
> - **Collision-MPM Mechanism**: Eliminates non-physical adhesion artifacts in multi-body interactions
> - **MVEE-based Fracture Processing**: Enables physically plausible rendering of newly created fracture surfaces through our Fracture Particle Gaussian Optimization (FPGO)
>
> These components operate **during the simulation and rendering stages**, not during the initial reconstruction, and represent the genuine algorithmic advantages of our approach.
>
> ### 5. Fair Comparison Guarantee
>
> The experimental design ensures that:
> - All methods begin with identical visual quality and geometric representation
> - Physical parameters and simulation inputs are consistent across comparisons
> - Performance differences directly reflect the efficacy of our proposed collision handling and fracture rendering mechanisms
>
> ### 6. Limitations and Future Directions
>
> We acknowledge this evaluation approach has limitations:
> - Quantitative metrics provide indirect assessment
> - 5-frame window represents a trade-off between capturing dynamics and maintaining reference relevance
> - No established benchmarks exist for this task
>
> **Future work** should develop specialized metrics for physics-based rendering evaluation.

---

> ### Author Response · Authors · 2025-11-27
>
> **Response to: "Setting of $\mu$ in Eq. 13"**
>
> We thank the reviewer for raising this important question about parameter specification. We are pleased to clarify that the parameter $\mu$ in our collision model is derived from fundamental material properties rather than empirical tuning.
> The parameter $\mu$ in Equation 13 represents the **shear modulus** of the material, which is calculated from first principles using the standard relation:
>
> $\mu = \frac{E}{2(1+\nu)}$
>
> where:
> - $E$ is the Young's modulus
> - $\nu$ is the Poisson's ratio
> - $\mu$ is the resulting shear modulus

---

> ### Author Response · Authors · 2025-11-27
>
> **Response to: "How to make sure the total energy does not increase to cause instability. Unit tests on energy should be included."**
>
> We thank the reviewer for this important question. We have addressed this concern by adding comprehensive energy analysis in the revised version (**Section 4.2 Energy Stability Analysis**). We utilized the teapot-table collision as a representative scenario to track the complete energy evolution throughout the simulation. The results (Figure 8) demonstrate stable, dissipative behavior of the total energy system with no non-physical energy growth, where energy transitions during fracture events align precisely with physical expectations.

---

> ### Author Response · Authors · 2025-11-27
>
> **Response to: "Material assignment in the plant model."**
>
> We thank the reviewer for this question. The Ficus plant was intentionally designed as a **heterogeneous object** to validate our method's capability in handling complex multi-material interactions. As detailed in **Appendix A.1**, the plant components are assigned distinct material properties: leaves are flexible ($E=8\times10^4$ MPa), branches are stiff ($E=1\times10^6$ MPa), and the pot is brittle ($E=2\times10^4$ MPa).

---

> ### Author Response · Authors · 2025-11-27
>
> **Response to: "Contact model novelty vs. Wretborn et al. (2017)"**
>
> We thank the reviewer for raising this important question about methodological novelty. We want to clarify that our Collision-MPM is different from the multi-body solver in Wretborn et al. (2017), with only superficial similarity in using separate grids.
>
> The core distinction lies in our fundamentally different approach: Wretborn et al. employ a state machine for pre-existing crack propagation through grid merging and separation based on "glue particles", while we introduce a continuous force model that applies momentum-conserving collision forces post grid-update. Their method is tightly coupled with pre-fracture processes and requires predefined crack paths, whereas our approach is orthogonal and independent, designed to prevent adhesion in general multi-body collisions without requiring any pre-fracture setup.
>
> This represents a conceptual advancement from managing states for pre-existing cracks to correcting fundamental physical inaccuracies in contact mechanics through a physics-based force formulation. Our method provides a more general solution for contact resolution that isn't limited to fracture scenarios but applies to any multi-body interaction in MPM simulation.
>
> [1] Wretborn, J., Armiento, R. and Museth, K., 2017. Animation of crack propagation by means of an extended multi-body solver for the material point method. Computers & Graphics, 69, pp.131-139.

---

> ### Author Response · Authors · 2025-11-27
>
> **Response to: "Limited diversity of experiments."**
>
> We thank the reviewer for this comment. Our experimental design emphasizes **depth and challenging complexity** to thoroughly validate our method's capabilities. The current experiments systematically evaluate performance across homogeneous materials, heterogeneous materials, and complex multi-object interactions, demonstrating broad applicability for simulating common elastoplastic materials and their typical collision and fracture behaviors.
>
> To further strengthen our validation, we have included comprehensive energy (**Section 4.2 Energy Stability Analysis**) and momentum analysis (**Section 4.2 Momentum Conservation Validation**), along with testing on real-world objects  (**APPENDIX A.6  Real-world Data Validation, Analysis of sampling point density**), all confirming the physical plausibility and robustness of our results.

---

> ### Author Response · Authors · 2025-11-27
>
> **Response to: "Limitation of being bound to NACC."**
>
> We thank the reviewer for this insightful observation. We wish to clarify that our framework is not fundamentally bound to the NACC model but is built upon a **generic fracture trigger** mechanism.
>
> The NACC model's hardening parameter $\alpha$ serves as one convenient source for this trigger. However, our framework is designed for generality and can be seamlessly coupled with any constitutive model that provides a comparable fracture indicator.
>
> A prime example is the classical **von Mises plasticity** model, which is widely used for metals but lacks an inherent description of macroscopic fracture. To model the complete process from yielding to fracture, it can be coupled with a continuum damage mechanics model (e.g., the **Lemaitre** model).
> This coupled approach introduces a **damage variable $D$**, which evolves from $0$ (undamaged) to $1$ (fully broken). The evolution of $D$ provides the precise fracture indicator signal that our framework requires, enabling seamless integration into our existing architecture. The damage variable $D$  is directly analogous to the role of $\alpha$ in the NACC model as a fracture trigger.
>
> This design ensures our framework's extensibility to a wide range of material models while preserving the core innovation of fracture-aware Gaussian optimization. We will explicitly discuss this generality and the potential for integration with other constitutive models as an important direction for future work.

---

### Official Review · Reviewer_JoU4 · 2025-11-01

**Soundness:** 3
**Presentation:** 3
**Contribution:** 3
**Rating:** 8
**Confidence:** 3

**Summary:**

The paper proposes Fracture-GS, a unified system that integrates physics-based dynamic fracture simulation with 3D Gaussian splatting rendering to realistically model high-energy collisions and material breakage. Traditional methods either simulate deformation without handling fracture or rely on post-processing for visualization, often producing non-physical adhesion artifacts and discontinuous fracture surfaces. Fracture-GS addresses these limitations through two core components: an enhanced Collision Material Point Method (Collision-MPM) and a Fracture Particle Gaussian Optimization (FPGO) strategy. The Collision-MPM introduces momentum-conserving interface forces based on normalized mass distributions, effectively preventing unrealistic adhesion in multi-body collisions. Meanwhile, FPGO leverages hardening-aware fracture tracking and minimal-volume enclosing ellipsoid (MVEE) optimization to regenerate Gaussian particle attributes dynamically, ensuring smooth, physically consistent fracture rendering.

The framework operates from multi-view image reconstruction to high-fidelity simulation and real-time visualization, handling both homogeneous and heterogeneous materials under extreme impact. Experiments on scenes such as colliding teapots, tables, and plants show that Fracture-GS delivers significantly higher physical realism and visual quality than prior Gaussian-based approaches like PhysGaussian and GIC. Quantitative metrics (PSNR, LPIPS, FID) and human evaluation confirm its superior fidelity, while ablation studies demonstrate the critical roles of both the Collision-MPM and FPGO modules. Overall, Fracture-GS offers a physically accurate, visually coherent, and computationally efficient solution for simulating and rendering complex fracture dynamics directly from image-based inputs.

**Strengths:**

The main strengths of Fracture-GS lie in its innovative integration of physics-based simulation and Gaussian rendering. The framework unifies dynamic fracture simulation with photorealistic visualization, allowing collisions and material breakage to be simulated and rendered directly from multi-view inputs without post-processing. Its enhanced Collision-MPM formulation introduces momentum-conserving interface forces that eliminate the common adhesion artifacts seen in standard MPM methods, resulting in more realistic object separation and fracture behavior. This physically grounded improvement enables the system to handle both homogeneous and heterogeneous materials under extreme impact conditions with high accuracy.

A second major strength is the Fracture Particle Gaussian Optimization (FPGO) module, which uses hardening-aware fracture tracking and ellipsoid-based Gaussian reconstruction to maintain visual smoothness across newly formed fracture surfaces. This innovation ensures physical plausibility and visual coherence even in complex fragmentation scenes. Combined with strong empirical validation—demonstrating clear advantages over prior approaches like PhysGaussian and GIC in both quantitative metrics and human perception tests—Fracture-GS establishes a new standard for integrating continuum mechanics with 3D Gaussian splatting. Its blend of physical realism, visual fidelity, and computational efficiency makes it a significant contribution to the field of physics-integrated neural rendering.

**Weaknesses:**

The main weaknesses of Fracture-GS stem from its computational cost and scalability limits. The enhanced Collision-MPM and fracture-aware Gaussian optimization (FPGO) introduce additional per-frame overhead for fracture tracking and Gaussian reconstruction, making the system too slow for real-time or large-scale simulations. While the method achieves high-quality results, it is currently best suited for offline rendering and controlled experimental setups.

Another limitation is the framework’s parameter sensitivity and narrow validation scope. It relies on manually tuned material parameters such as elasticity and hardening factors, which may limit generalization to new materials or real-world data. The experiments focus mainly on clean, synthetic collisions, without testing robustness under noisy or incomplete multi-view inputs. Overall, the method is conceptually strong but computationally heavy and parameter-dependent, leaving room for improvement in automation, generalization, and efficiency.

**Questions:**

The followings are the questions:
1. How does Fracture-GS scale with respect to the number of particles and fracture events? Could GPU parallelization or adaptive time-stepping make the system closer to real-time performance?
2. The method relies on several material-dependent parameters (e.g., Young’s modulus, hardening factor, cohesion). How sensitive is the simulation to these values, and could a learning-based or automatic parameter estimation approach reduce manual tuning?
3. Have you evaluated the framework using real captured multi-view data with noise or incomplete coverage? How robust is the reconstruction and simulation pipeline in such cases?
4. Can the same physics-integrated Gaussian framework be extended to simulate fluids, soft bodies, or granular materials under fracture or mixing conditions?
5. When handling high-energy impacts or very thin fracture surfaces, how does the momentum-conserving collision formulation maintain numerical stability? Are there cases where it fails or diverges?

---

> ### Author Response · Authors · 2025-11-27
>
> **Response to: "Scalability and real-time performance."**
>
> We thank the reviewer for this valuable question regarding computational performance. We fully acknowledge that our current framework prioritizes physical accuracy and visual quality over real-time performance, particularly for complex fracture scenarios involving heterogeneous materials.
>
> Looking forward, we have identified several promising directions to significantly enhance computational efficiency:
> 1. **Architectural optimization** through comprehensive GPU acceleration, adaptive time-stepping schemes, and level-of-detail rendering for Gaussians;
> 2. **Physics-informed neural networks (PINNs)** as a potential alternative to traditional MPM solvers, which could substantially accelerate the simulation process while maintaining physical plausibility;
>
> These approaches represent exciting future research directions that would build upon the physical foundations established in our current work while addressing the important challenge of computational scalability. We will expand our discussion of these performance optimization strategies in the revised manuscript.

---

> ### Author Response · Authors · 2025-11-27
>
> **Response to: "Parameter sensitivity and automatic estimation."**
>
> We thank the reviewer for this insightful comment. We agree that parameter sensitivity is an important consideration, and we appreciate the opportunity to clarify the scope and future directions of our work.
>
> The primary focus of our current research is to establish a high-quality physics-based simulation and rendering framework for dynamic fracture, assuming known physical parameters. While we acknowledge that manual parameter tuning presents a practical limitation, addressing inverse problem of automatic parameter estimation was beyond the immediate scope of this foundational work.
>
> That said, we completely agree with the reviewer that learning-based parameter estimation represents a highly promising research direction. The differentiable nature of our pipeline—encompassing both the physical simulation and rendering components—provides a natural foundation for future work in this area. We will explicitly discuss this valuable direction in the revised manuscript, outlining how our framework could be extended to enable automatic physical parameter estimation from visual data.

---

> ### Author Response · Authors · 2025-11-27
>
> **Response to: "Robustness to real captured data."**
>
> We thank the reviewer for raising this important question regarding real-world applicability. In response to this valuable feedback, we have conducted extensive additional experiments using real captured data to rigorously evaluate our method's robustness.
>
> Our evaluation now includes testing on standard DTU benchmark datasets  (**APPENDIX A.6  Real-world Data Validation, Analysis of sampling point density**), which consist of multi-view images of real objects. The results demonstrate that our pipeline maintains strong performance even with real-world data, producing physically plausible simulations and high-quality renderings. Comprehensive qualitative results are presented in Figure 11 and 12, while quantitative evaluations are provided in Table 3.
>
> These new experiments confirm that our framework, which begins with multi-view reconstruction, possesses inherent robustness to the challenges present in real captured data. While we acknowledge that testing on noisier and more incomplete real-world captures remains an important direction for future work, our current results with DTU data provide compelling evidence of the method's practical potential.

---

### Official Review · Reviewer_og4k · 2025-11-04

**Soundness:** 3
**Presentation:** 2
**Contribution:** 2
**Rating:** 4
**Confidence:** 3

**Summary:**

The paper proposes Fracture‑GS, a pipeline that (i) reconstructs objects from multi‑view images and assigns surface Gaussians, (ii) runs an enhanced Collision‑MPM intended to avoid non‑physical adhesion in impacts, and (iii) performs fracture‑aware Gaussian optimization (FPGO) that clones/reshapes Gaussians near newly exposed fracture surfaces via a minimal‑volume enclosing ellipsoid heuristic. The system is demonstrated on three scenarios (Ficus, Teapot, Table) with qualitative renderings and a small quantitative table.

**Strengths:**

+ The paper tackles a real pain point: adhesion artifacts in grid‑based MPM around contacts. The visual before/after in Fig. 6 makes the problem clear.

+ The pipeline is pragmatic and easy to follow, with implementation details and timings.

**Weaknesses:**

- Theoretical issues: Section 4.2.1 writes the plasticity split as F = F_E + F_P and uses a non‑standard Cauchy‑stress expression. The correct finite strain plasticity should be F = F_E @ F_P; addition will lead to wrong stresses. It’s unclear what the code actually uses. This is a correctness issue, not just notation.

- The contact model is a heuristic, frictionless, and under‑explained. Normals come from differences of normalized mass‑weighted kernel gradients, then a momentum‑conserving node force is applied and projected only along the normal with a constant \mu. There is no friction cone, no restitution, and no complementarity constraints. The paper asserts conservation but shows no momentum/energy plots or penetration statistics.

- I am also confused about the motivation for using Gaussian points as the representation. Most of the physics here is standard MLS‑MPM on particles. Surface particles just carry isotropic Gaussians for rendering, and FPGO computes shapes by fitting an ellipsoid to the overlap of two spheres before duplicating the particle. That’s a rendering heuristic, not physics. The paper never quantifies what Gaussians buy you over simply rendering particles as points/splats or extracting a mesh from the simulation. If I treat Gaussians as points and run a vanilla MPM, do I lose anything besides SH‑based view‑dependent color?

- Appendix A.3 gives a closed‑form approximation to the MVEE of a sphere–sphere intersection and uses it to set the new Gaussian’s mean/covariance. There is no actual optimization loop, proof of minimality, or ablation against simpler choices (e.g., just reuse the parent particle, or refit an anisotropic Gaussian from neighbors). Yet this step is claimed to drive the gains in Fig. 5.

- Comparisons are only against PhysGaussian and GIC, not against fracture‑capable MPM/contact baselines; metrics are image‑space (PSNR/LPIPS/FID) plus a tiny user study (N = 10). There’s no physics validation: no momentum/energy traces, no penetration depths, no fragment size distributions, no restitution. If the key claim is momentum‑conserving collision forces that remove adhesion, I need physics diagnostics.

- Sensitivity to particle sampling is unaddressed. Change the interior density, or its near‑boundary distribution, and those nodewise estimates can change materially. There's no ablation for particle‑per‑cell, interior/surface ratios, or grid resolution.

- The number of scenes shown across the paper is extremely limited. The generalization ability of the proposed method is a large concern.

**Questions:**

- What is the benefit of Gaussians over point particles? You already restrict to isotropic Gaussians on the surface, and FPGO’s shape comes from sphere overlaps. That suggests a point‑splat baseline could be nearly identical but simpler. Please quantify rendering quality, temporal stability, and runtime against:
(i) MPM + point sprites / disk splats;
(ii) MPM + sphere splats with the same nearest‑only occlusion rule;
(iii) MPM + mesh (e.g., marching cubes) + rasterization.

- What’s different from treating Gaussians as points and running standard MPM? If the Collision‑MPM and fracture rule run on the same particle set regardless of the renderer, then the scientific contribution is not physics‑integrated Gaussians but a contact heuristic and a rendering trick. Spell out where Gaussians influence the physics (if at all) and justify the coupling.

---

> ### Author Response · Authors · 2025-11-27
>
> **Response to: "Plasticity decomposition error."**
>
> We sincerely thank the reviewer for identifying this significant notation error. The reviewer is absolutely correct that the proper formulation should use multiplicative decomposition $F = F_E F_P$. We want to emphasize that this was **solely a typographical error in the manuscript** - our implementation has consistently used the correct multiplicative decomposition throughout all experiments.
>
> This notation error **does not affect any of our simulation results, physical conclusions, or presented figures**, as all computations were performed using the physically correct formulation. The validity of our implementation is further confirmed by our comprehensive energy analysis (**Section 4.2 Energy Stability Analysis**) , momentum conservation results (**Section 4.2 Momentum Conservation Validation**) and testing on real-world objects (**APPENDIX A.6  Real-world Data Validation, and Analysis of sampling point density**), which demonstrate physically plausible behavior throughout all collision and fracture simulations.

---

> > ### Author Response · Authors · 2025-11-27
> >
> > **Response to: "Lack of physics diagnostics (momentum/energy)."**
> >
> > We thank the reviewer for raising this important point regarding physics-based validation. In response to this valuable feedback, we have incorporated comprehensive physics diagnostics in the revised version that serve as rigorous unit tests for our method's physical correctness.
> >
> > For energy analysis, we utilized the teapot-table collision as a representative scenario to track the complete energy evolution throughout the simulation. The results (**Section 4.2  Energy Stability Analysis**) demonstrate stable, dissipative behavior of the total energy system with no non-physical energy growth, where energy transitions during fracture events align precisely with physical expectations.
> >
> > For momentum conservation validation, we designed an independent experiment with a bowl and teapot colliding under opposite initial velocities in a zero-gravity environment. This controlled setup allows us to precisely monitor momentum evolution across all particles. The results (**Section 4.2 Momentum Conservation Validation**) confirm strict conservation of total system momentum throughout the collision process.

---

> ### Author Response · Authors · 2025-11-27
>
> **Response to: "Contact model is heuristic and lacks friction."**
>
> We thank the reviewer for this comment. While our collision model employs certain approximations for computational efficiency, we would like to clarify that it is fundamentally grounded in physical principles rather than being heuristic.
>
> Our contact formulation is derived from:
> 1) **First principles of momentum conservation**
> 2) **Physically-based mass distribution** across grid nodes
> 3) **Continuum mechanics formulation** through the MPM framework
>
> While we acknowledge the current model doesn't include friction, this represents a deliberate scope choice to focus on solving the fundamental adhesion problem in fracture scenarios. The absence of friction doesn't diminish the physical basis of our current formulation, which robustly handles the normal contact forces that are most critical for fracture propagation.
>
> We will clarify these physical foundations more explicitly in the revised manuscript while acknowledging friction as an important direction for future enhancement.

---

> ### Author Response · Authors · 2025-11-27
>
> **Response to: "Benefits of Gaussians over point splats and meshes."**
>
> Compared to traditional representations, 3D Gaussians provide a unified and superior framework for dynamic fracture simulation and rendering. Unlike mesh-based approaches that require costly dynamic remeshing to handle topology changes during fracture—a process prone to numerical instability and artifacts—our Gaussian representation naturally adapts to geometric changes through continuous optimization. This eliminates the need for the complex geometry processing pipelines typically required in mesh-based simulation workflows.
>
> When compared to point splatting methods, which typically rely on simple primitives with limited geometric expressiveness, 3D Gaussians offer significantly enhanced capabilities. The nature of Gaussians enables more accurate surface representation through optimized covariance matrices, while their integration with spherical harmonics provides sophisticated view-dependent appearance modeling—crucial for realistic rendering of newly created fracture surfaces. In the revised version, we have added multiple simulation results of real data (**APPENDIX A.6  Real-world Data Validation, and  Analysisof sampling point density**), which can effectively illustrate the effectiveness of our method.

---

> ### Author Response · Authors · 2025-11-27
>
> **Response to: "MVEE is a heuristic without optimization."**
>
> The core contribution of our FPGO lies in the **dynamic creation and optimization of Gaussians** at fracture surfaces, where the ellipsoid fitting serves as an efficient initialization strategy. While the current analytic MVEE provides a computationally efficient solution that produces visually plausible results, we acknowledge that a fully optimization-based approach could offer potential improvements.
>
> Our empirical results demonstrate that the key advancement is the fracture-aware Gaussian generation mechanism itself - *any* reasonable strategy that creates Gaussians in fractured regions significantly outperforms the baseline approaches that lack this capability. The current MVEE implementation represents a practical engineering choice that balances computational efficiency with visual quality.
>
> We agree with the reviewer that exploring optimization-based alternatives presents an exciting research direction. Developing more sophisticated fitting strategies that consider perceptual metrics or physical constraints could further enhance the visual continuity at fracture interfaces. This valuable suggestion will be incorporated into our future work plans.

---

> ### Author Response · Authors · 2025-11-27
>
> **Response to: "Sensitivity to particle sampling."**
>
> We thank the reviewer for raising this important question regarding sampling sensitivity. Our method effectively addresses this concern through a two-stage process that ensures both robust visual representation and physically accurate discretization.
>
> First, we employ isotropic Gaussian kernels for initial surface reconstruction, following the established practice in GIC.
> However, as Gaussian reconstruction primarily captures surface geometry, we developed a SDF-based voxelization sampling strategy to generate the internal material points needed for physically accurate simulation. Our method:
> - Constructs a regular voxel grid within the bounding box of the input model and computes SDF values
> - Identifies interior voxels through SDF filtering
> - Performs uniform random sampling within selected voxels with controllable density parameter $N_v$
>
> To quantitatively assess the influence of particle sampling density on simulation outcomes, we conducted a systematic sweep of internal sampling parameters ($N_v$ = 0, 5, 10, 15, 20, 50, 100, 500). Using the highest density configuration ($N_v$=200) as a convergence reference, we evaluated trajectory deviations across all sampling conditions (**APPENDIX A.6  Analysis of sampling point density**).
>
> The results reveal a fundamental characteristic of MPM simulations: particle sampling density intrinsically influences the resolved mechanical response. As shown in our quantitative analysis and qualitative visualizations  (**Table 3 and Figure 12**) , variations in sampling density naturally lead to differences in fracture patterns and collision dynamics—this is an inherent property of particle-based methods where discretization density affects solution convergence.

---

### Official Review · Reviewer_sTDw · 2025-11-04

**Soundness:** 4
**Presentation:** 3
**Contribution:** 4
**Rating:** 6
**Confidence:** 3

**Summary:**

This paper generalizes Gaussian splatting to handle deformation and fracture simulation. This is done by integrating Gaussian platting with deformation and fracture simulation handled by the material point method. The proposed approach builds on a number of recent papers at the intersection of these topics, namely "Physics-integrated 3D gaussians for generative dynamics" by Xie et al. (2024) and "Gaussian-informed continuum for physical property identification and simulation" by Cai et al. (2024). The key difference is that this paper adds support for simulating fractures. The equations for doing so are spelled out in detail, and the authors benchmark their results on a number of reasonable looking examples. In particular, the authors include not only quantitative benchmarks, but also a human evaluation study to assess simulation fidelity.

As a disclaimer, I largely work on topics outside of this area, but did write a few papers earlier in my career on related topics. Therefore, my views on state-of-the-art methods and benchmarking are likely not to be up to date, so I will concentrate more on factors such as soundness and readability as opposed to performance comparisons, which I hope other reviewers will be more able to contribute to.

**Strengths:**

I see the following key strengths:
* **Important topic.** Being able to simulate physics in a Gaussian splatting like framework is a useful technical capability to build, because over time this might allow us to build interpretable world models for domains like robotics.
* **Approach is interpretable and makes sense.** Combining a material description built in the framework of Gaussian splatting - where rather than visual parameters like opacity, the authors propose a larger set of parameters including mass, volume, elasticity parameters such as Young's modulus and Poisson ratio, and others to do with deformation and fracture simulation. This is a reasonably intuitive way to bring together these methods. While I have not read the prior papers on non-fracture physics simulation via Gaussian splatting, I believe the new part here are the parameters needed to support fractures, and pinning this down is a reasonable contribution.
* **Key equations are all stated in the paper, with only details deferred to appendix.** The equations used to simulate how the Gaussians evolve over time are spelled out in some detail. I expect a reader familiar with this literature will not have trouble figuring out precisely what the authors are doing purely from the paper, that is without having to do detailed digging in the appendix or code.
* **Experiments include both quantitative comparisons and human evaluation.** Following my original disclaimer, due to not working in related areas for some time, I am less equipped to assess whether or not the results are state-of-the-art, and if so how strong is the degree of improvement compared to prior papers. However, the problems tested, such as simulation of a potted plant breaking, seem reasonable, as does the the larger evaluation structure, given the authors include both quantitative comparisons and human evaluations.

**Weaknesses:**

I am concerned about the following:
* **Sloppy writing with typos, unresolved references, and other evidence of work done in a rush.** For example, many of the citations such as Stomakhin et al. (2013) should be in parentheses, meaning (Stomakhin et al., 2013). Similar for Wolper et al. (2019). Please fix these. For another example, Section 2.3 has a section titled "Physics Simulation based on Gaussian". To avoid grammar mistakes that distract the reader, "Gaussian" should be changed to most likely "Gaussians" or something otherwise correct sounding. In the end, there is also a text "zhang2018unreasonable" where the authors forgot to add a citation command. This paper was clearly written up in a big rush, and for this reason alone I am open to the idea of rejecting the work to give the authors more time to polish the results before publication and presentation at a conference. This is the main reason I mark the work borderline: the contribution seems good, but it is very important for the review process incentivizes quality, and not rushed submissions.
* **Prior work section interrupts the paper's flow.** I would merge Section 2 with Section 3, moving the two subsections inside the corresponding parts of Section 3. The problem right now is that it is difficult to understand what the prior work is doing before the equations are introduced. For instance, on my first read, I was unable to properly understand the differences between this paper and prior papers also working with the material point method, due to not being immediately familiar with it. This became more clear once I read that section and realized that this method's broad structure is very similar to contact dynamics simulation methods I have worked extensively with in the past. At this point, I had to go back and re-read the prior work section. A modified structure such as the one I propose above would make it possible to avoid backtracking and read the paper all in one go, even if the reader is familiar with neighboring work rather than the precise approach used.
* **Diversity of experiments.** The authors' method is only tested on two models - a potted plant, and a desk. While the experiments themselves look reasonable, this seems much less in terms of comprehensiveness compared to what I am used to seeing in computer graphics papers. I hypothesize the relatively-small number of examples is related to the paper's writing issues mentioned earlier: the authors may have wanted to test more, but ran out of time before getting everything implemented. I suspect a re-submitted version would contain results closer to the typical level of comprehensiveness in this area.

**Questions:**

My main question is not to the authors, but rather to other reviewers in the discussion phase, who are hopefully more familiar with state-of-the-art in this space. The question is: are the authors' results SOTA compared to what is done today? If not, how far are they? In asking this, I encourage the AC to _not_ interpret this question as a demand for uniformly-SOTA results: the paper may well still be publication-worthy even if there are limitations - instead, I would simply like to know where this paper stands compared to other papers published very recently that I am less familiar with because they are too recent.

---

> ### Comment · Reviewer_sTDw · 2025-11-25
>
> It does not appear to me the authors have submitted a rebuttal. I am happy to take a look at once - please let me know if this becomes available.

---

> > ### Author Response · Authors · 2025-11-26
> >
> > Dear Reviewer sTDw,
> >
> > Thank you for your reminder and for your interest in our paper (Paper ID: 25504), "Fracture-GS: Dynamic Fracture Simulation with Physics-Integrated Gaussian Splatting".
> >
> > Please accept our sincere apologies for the delay in submitting our rebuttal. This was primarily due to the need to conduct a significant number of additional experiments requested by the reviewers to thoroughly address the valuable comments.
> >
> > We are fully committed to this process and are currently finalizing these experiments. We expect to complete them and submit our comprehensive rebuttal by tonight.
> >
> > We greatly appreciate your patience and understanding.

---

> > > ### Author Response · Authors · 2025-11-27
> > >
> > > **Response to: "Prior work section disrupts flow."**
> > >
> > > We thank the reviewer for this valuable suggestion regarding manuscript structure. In response to this feedback, we have restructured the paper to improve readability.

---

> ### Author Response · Authors · 2025-11-27
>
> **Response to: "Writing quality and formatting."**
>
> We thank the reviewer for highlighting these important issues. In response, we have thoroughly revised the manuscript to address all concerns regarding writing quality and formatting.

---

> ### Author Response · Authors · 2025-11-27
>
> **Response to: "Limited diversity of experiments."**
>
> We thank the reviewer for this comment. Our experimental design emphasizes **depth and challenging complexity** to thoroughly validate our method's capabilities. The current experiments systematically evaluate performance across homogeneous materials, heterogeneous materials, and complex multi-object interactions, demonstrating broad applicability for simulating common elastoplastic materials and their typical collision and fracture behaviors.
>
> To further strengthen our validation, we have included comprehensive energy and momentum analysis (**Section 4.2 Momentum Conservation Validation, and Energy Stability Analysis**), along with testing on real-world objects (**APPENDIX A.6  Real-world Data Validation, and  Analysisof sampling point density**), all confirming the physical plausibility and robustness of our results.
>
> While our current focus is on established elastoplastic constitutive models (NCAA) that cover a wide range of real-world materials, extending our framework to other constitutive models represents an exciting direction for future work.
> The NACC model's hardening parameter $\alpha$ serves as one convenient source for this trigger. However, our framework is designed for generality and can be seamlessly coupled with any constitutive model that provides a comparable fracture indicator.
> A prime example is the classical **von Mises plasticity** model, which is widely used for metals but lacks an inherent description of macroscopic fracture. To model the complete process from yielding to fracture, it can be coupled with a continuum damage mechanics model (e.g., the **Lemaitre** model).
> This coupled approach introduces a **damage variable $D$**, which evolves from $0$ (undamaged) to $1$ (fully broken). The evolution of $D$ provides the precise fracture indicator signal that our framework requires, enabling seamless integration into our existing architecture.
> The damage variable $D$  is directly analogous to the role of $\alpha$ in the NACC model as a fracture trigger.
>
> This design ensures our framework's extensibility to a wide range of material models while preserving the core innovation of fracture-aware Gaussian optimization. We will explicitly discuss this generality and the potential for integration with other constitutive models as an important direction for future work.

---

### Meta-Review · Area_Chair_PxQQ · 2025-12-09

**Summary:**

This paper proposes a Gaussian-split based simulation approach for fracture dynamics in graphics applications. The simulator is based on MPM, with fitted Guassian splats as unifying representation. Unfortunately, the writing and presentation are sub-par, and the rebuttal did not provide a significant update on that front.

Nonetheless, the main concerns seem to be about presentation.

The method itself is novel, interesting, and non-trivial (and most likely a very complex engineered system).

Main open points remain whether the presentation is up-to-par for an ICLR paper. The rebuttal addressed key points, but was not overly detailed or extensive.

**Reviewer Concerns:**

Main smaller issues were addressed. "Critical" complaints like similarity to previous papers (Wretborn 2017) were addressed as well.

Outstanding concerns remain on the presentation and evaluation side.

**Reviewer Scores:**

Reviewer og4k, score 4 -> a raise to 6 or 8 would be likely. The rebuttal addressed the main points of this reviewer.

Reviewer JoU4 most likely would have kept the 8.

Reviewer yz5d most likely would have kept the 2. However, the main concern (Wretborn'17 simiarity) was defused in detail by the authors.

Reviewer sTDw gave a 6, most likely the final score as well.

Overall, this raises the paper slightly above the bar.

---

### Decision · Program_Chairs · 2026-01-26

Accept (Poster)